Relative contribution of ecological and biological attributes in the fine-grain structure of ant-plant networks

Díaz-Castelazo Cecilia diazcastelazogm@gmail.com 1
Martínez-Adriano Cristian A. 1 2
Dáttilo Wesley 3
Rico-Gray Victor 4
1 Red de Interacciones Multitróficas, Instituto de Ecología, A.C. , Xalapa , Veracruz , México
2 Facultad de Ciencias Forestales, Universidad Autónoma de Nuevo León , Linares , Nuevo León , México
3 Red de Ecoetología, Instituto de Ecología, A.C. , Xalapa , Veracruz , México
4 Instituto de Neuroetología, Universidad Veracruzana , Xalapa , Veracruz , México
Huber Dezene
Electronic publication date: 2020 Feb 28
Publication date: 2020
Volume: 8
Electronic Location ID: e8314
Received 2015 Nov 10; Accepted 2019 Nov 28
Copyright: ©2020 Díaz-Castelazo et al.
Copyright year: 2020
Copyright holder: Díaz-Castelazo et al.
License: This is an open access article distributed under the terms of the Creative Commons Attribution License, which permits unrestricted use, distribution, reproduction and adaptation in any medium and for any purpose provided that it is properly attributed. For attribution, the original author(s), title, publication source (PeerJ) and either DOI or URL of the article must be cited.
License URL: https://creativecommons.org/licenses/by/4.0/

Keywords: Ant behavior, Ant-plant mutualism, Biological attributes, Community ecology, Determinants of network structure, Ecological networks, Extrafloral nectaries, Habitat structure, Invasive species

Funding: CONACYT 118946 Instituto de Ecología, A.C. (INECOL) 902-16 2003011143 The authors received funding from the following grants and institutions: Grant number 118946 for Cecilia Díaz-Castelazo by CONACYT. Project number 902-16 for Victor Rico-Gray and Project number 2003011143 for Cecilia Díaz-Castelazo, both by Instituto de Ecología, A.C. (INECOL). The funders had no role in study design, data collection and analysis, decision to publish, or preparation of the manuscript.

==============================
Background

Ecological communities of interacting species analyzed as complex networks have shown that species dependence on their counterparts is more complex than expected at random. As for other potentially mutualistic interactions, ant-plant networks mediated by extrafloral nectar show a nested (asymmetric) structure with a core of generalist species dominating the interaction pattern. Proposed factors structuring ecological networks include encounter probability (e.g., species abundances and habitat heterogeneity), behavior, phylogeny, and body size. While the importance of underlying factors that influence the structure of ant-plant networks have been separately explored, the simultaneous contribution of several biological and ecological attributes inherent to the species, guild or habitat level has not been addressed.

Methods

For a tropical seasonal site we recorded (in 48 censuses) the frequency of pairwise ant-plant interactions mediated by extrafloral nectaries (EFN) on different habitats and studied the resultant network structure. We addressed for the first time the role of mechanistic versus neutral determinants at the ‘fine-grain’ structure (pairwise interactions) of ant-plant networks. We explore the simultaneous contribution of several attributes of plant and ant species (i.e., EFN abundance and distribution, ant head length, behavioral dominance and invasive status), and habitat attributes (i.e., vegetation structure) in prevailing interactions as well as in overall network topology (community).

Results

Our studied network was highly-nested and non-modular, with core species having high species strengths (higher strength values for ants than plants) and low specialization. Plants had higher dependences on ants than vice versa. We found that habitat heterogeneity in vegetation structure (open vs. shaded habitats) was the main factor explaining network and fine-grain structure, with no evidence of neutral (abundance) effects.

Discussion

Core ant species are relevant to most plants species at the network showing adaptations to nectar consumption and deterrent behavior. Thus larger ants interact with more plant species which, together with higher dependence of plants on ants, suggests potential biotic defense at a community scale. In our study site, heterogeneity in the ant-plant interactions among habitats is so prevalent that it emerges at community-level structural properties. High frequency of morphologically diverse and temporarily-active EFNs in all habitats suggests the relevance and seasonality of plant biotic defense provided by ants. The robust survey of ecological interactions and their biological/ecological correlates that we addressed provides insight of the interplay between adaptive-value traits and neutral effects in ecological networks.

Introduction

The interactions among species occurring at a community have been studied recently with a complex network perspective, where interacting species (i.e., plants and animals) are graphically represented as nodes and their trophic interactions as links (Bascompte et al., 2003). Such studies have paid important attention to network structure and its underlying factors, both for mutualistic or antagonistic interactions (Bascompte & Jordano, 2007; Vázquez, Chacoff & Cagnolo, 2009; Díaz-Castelazo et al., 2013; López-Carretero et al., 2014). Unraveling how interactions among species are structured in communities or ecosystems is crucial for understanding the ecological and evolutionary processes that support ecosystem function and diversity (Herrera & Pellmyr, 2009). Furthermore, understanding the architecture of species relationships may help predict how ecosystems respond either to abiotic or human-derived changes (Bascompte, 2010).

Ecological network studies have shown that interactions among species are frequently asymmetric and species dependence on their counterparts is more complex than expected at random (Bascompte, Jordano & Olesen, 2006; Guimarães et al., 2007). For networks of mutualistic interactions a nonrandom “nested” structure is frequently observed, where more specialist species tend to interact with specific subsets of those species interacting with the more generalist species (Almeida-Neto et al., 2008; Bascompte, 2010). Thus, nested structure implies that interactions occur asymmetrically in a group of generalist species that comprise most interactions in the network (network core) (Dáttilo, Guimarães & Izzo, 2013b) and a group of specialist species that maintain few interactions mostly or exclusively with generalist species. Such as a nested architecture has been found to increase network robustness against loss of species (Memmott, Waser & Price, 2004; Bascompte, Jordano & Olesen, 2006) and to maximize the number of coexisting species supported by these networks (Bastolla et al., 2009; Thébault & Fontaine, 2010). A nested pattern of links in mutualistic interaction networks could result from several ecological and evolutionary processes: for instance, the complementarity and convergence of phenotypic traits between both sets of interacting species (Thompson, 2005; Stang et al., 2006; Stang, 2007; Rezende, Jordano & Bascompte, 2007).

Another nonrandom structural pattern in ecological interaction networks is the ‘modularity’ or ‘compartmentalization’, which is characterized by a group of species interacting more strongly among themselves than with other species or subsets in the network (Bascompte, 2010). The modular pattern is more frequently observed for networks of antagonistic interactions (Cagnolo, Salvo & Valladares, 2011). Like nestedness in mutualistic networks, modularity is thought to increase the persistence of species in antagonistic networks (Thébault & Fontaine, 2010).

Proposed mechanisms affecting overall network structure are diverse, including habitat heterogeneity constraints (Pimm & Lawton, 1980; López-Carretero et al., 2014), phylogeny (Rezende et al., 2007; Cagnolo, Salvo & Valladares, 2011), body size (Cohen et al., 2005; Rezende, Jordano & Bascompte, 2007; Chamberlain & Holland, 2009) encounter probability based on natural abundance of species (Vázquez, Chacoff & Cagnolo, 2009; Blüthgen, 2010; Dáttilo et al., 2014a), and variation in spatiotemporal co-occurrence (Rico-Gray et al., 2012; Sánchez-Galván, Díaz-Castelazo & Rico-Gray, 2012; Díaz-Castelazo et al., 2013; Junker et al., 2013; López-Carretero et al., 2014). Environmental changes may alter network structure and thus, favor evolutionary responses in opposing directions for different species (Guimarães, Jordano & Thompson, 2011). The reorganization of network structure due to the indirect effects of coevolution, may explain why and how mutualisms persist amid the turnover of species and interactions through space and time (Guimarães et al., 2017).

Several aggregate network properties such as nestedness, connectance (the proportion of realized interactions from all the ones possible given the number of species), and interaction asymmetry (i.e., asymmetry of dependence of plants on animals and vice versa) may also emerge due to properties inherent to communities (Jordano, Bascompte & Olesen, 2003; Jordano, Bascompte & Olesen, 2006). These causes include the different abundances of species, community sampling biases (that affect the detectability of some interactions), and the spatio-temporal overlap of species (i.e., co-occurrence) (Vázquez et al., 2007; Vázquez et al., 2009; Blüthgen et al., 2008).

Although relative species abundance (Vázquez et al., 2009; Dáttilo et al., 2014a) and spatio-temporal overlap—considered as ‘neutral’ causes of network structure—could explain overall network structure, they fall short of predicting the frequency of pairwise interactions (Vázquez et al., 2009; Poisot, Stouffer & Gravel, 2015). Indeed, the frequency of interactions occurring for any given pair of species within the network can vary significantly even if the overall network topology remains the same (Vázquez, Morris & Jordano, 2005; Vázquez et al., 2007; Vázquez et al., 2009; Díaz-Castelazo et al., 2010; Dáttilo et al., 2014d; Sánchez-Galván, Díaz-Castelazo & Rico-Gray, 2012).

The pattern and frequency of pairwise interactions (Bascompte, 2009) is what we refer to here as, the ‘fine-grain’ structure of the network, and is relevant since it could potentially demonstrate convergence or complementarity between species (Thompson, 2005; Guimarães, Jordano & Thompson, 2011). Thus, a current challenge in ecological network studies is to infer which processes are involved in the structuring the fine-scale patterns of interaction networks and how these may change over time (Ramos-Robles, Andresen & Díaz-Castelazo, 2016). Temporal changes in network structure and species composition may occur because of seasonal variability in weather (Rico-Gray et al., 2012), food abundance (Carnicer, Jordano & Melián, 2009; López-Carretero et al., 2014; Ramos-Robles, Andresen & Díaz-Castelazo, 2016), or plant traits (López-Carretero, et al., 2016). Progress in understanding the determinants of network patterns requires datasets with detailed information of natural history such as spatial or temporal variation, morphological, behavioral, or life-history traits, which explain interspecific differences observed between species in the number and strength of interactions (Stang et al., 2006; Carnicer, Jordano & Melián, 2009; Junker et al., 2013; (López-Carretero, et al., 2016)).

The study of ant-plant interactions at a community level has also been approached with the theoretical/analytical framework of ecological networks (Chamberlain & Holland, 2009; Díaz-Castelazo et al., 2010; Dáttilo et al., 2013a; Fagundes et al., 2017). These ant-plant interactions are mediated by several plant rewards for ants, such as extrafloral nectar, food bodies, fleshy diaspores, or plant domatia (Rico-Gray & Oliveira, 2007). At a community-level, plants providing good-quality extrafloral nectar are highly attractive to ants and accumulate more interactions with aggressive and territorial ant species (Blüthgen & Fiedler, 2004; Dáttilo, Díaz-Castelazo & Rico-Gray, 2014c), but more importantly, ant visits may result in a reduced herbivory damage (Oliveira et al., 1999; Cuautle & Rico-Gray, 2003; Fagundes et al., 2017). Plenty of variation in extrafloral nectaries (EFN thereafter) attributes exist, including nectar volume, the amount of secreted sugar, variable gland size and morphology, position of EFNs within plant organs, and differential attractiveness to ant foragers (Koptur, 1992; Wäckers & Bonifay, 2004; Díaz-Castelazo, Chavarro-Rodríguez & Rico-Gray, 2017). Many of these attributes show phenotypic plasticity or context-dependency (Koptur, 1992; Rudgers, 2004; Wäckers & Bonifay, 2004). In this context, plant investment in nectar production and quality is a very important factor modifying the benefit received by plants through biotic defense, and fitness-related outcomes of the interaction (Rudgers & Gardener, 2004; Holland, Chamberlain & Horn, 2009).

Ant-plant networks, including potentially mutualistic interactions (‘potentially’, because benefits were assessed only for few interactions, see: Horvitz & Schemske, 1984; Rico-Gray et al., 1989; Cuautle, Rico-Gray & Díaz-Castelazo, 2005; Rico-Gray & Oliveira, 2007), have been recently addressed focusing on their spatio-temporal variation (Díaz-Castelazo et al., 2010; Sánchez-Galván, Díaz-Castelazo & Rico-Gray, 2012; Díaz-Castelazo et al., 2013; Dáttilo, Guimarães & Izzo, 2013b; Dáttilo et al., 2014b) and/or determining biotic/abiotic factors; among the latter, temperature and precipitation (Rico-Gray et al., 2012), soil pH (Dáttilo et al., 2013a), and the temporal variation in the percentage of plants with active extrafloral nectaries that mediate these interactions (Lange, Dáttilo & Del-Claro, 2013), have important effects on the structure (i.e., nestedness, specialization) of ant–plant networks mediated by extrafloral nectaries (EFNs).

Some studies have shown that variation in abundance of ants among different types of vegetation, partially explains the network structure of mutualistic interactions, where abundant ant species usually interact with more plant species (Dáttilo et al., 2014b). Similarly, the abundance of plants-bearing extrafloral nectaries and plant size (Lange, Dáttilo & Del-Claro, 2013) are important predictors of asymmetric (i.e., nested) interactions between plants and ants in ant-plant networks. Ant species attributes may influence the structure in ant-plant networks, include the social recruitment behavior of ants (Dáttilo et al., 2014b), as well as its invasive potential (Ness & Bronstein, 2004). Once a worker ant forager finds a profitable food source (i.e., extrafloral nectar) it will (or not if it is a solitary forager) recruit nestmates using variable strategies (Dornhaus & Powell, 2010). These strategies includes group recruitment, tandem running, mass recruitment by pheromone trail, trunk trails, team transport, etc. which are highly variable depending on ant species/genus or ecological context (Ness & Bronstein, 2004; Lach & Hooper-Bui, 2010). Often, aggressive behavior of ants and numerical dominance are attributes that influence the recruitment and competition abilities (Parr & Gibb, 2010). Ant dominance hierarchy determined by ant behavior also influences network structure since ant species found in the central core of the network are frequently competitively superior (i.e., showing massive recruitment and resource domination), compared with peripheral species with fewer interactions (Dáttilo, Díaz-Castelazo & Rico-Gray, 2014c). Furthermore, invasive ant species, given their opportunism, recruitment behavior, and numeric dominance (Ness & Bronstein, 2004; Lach & Hooper-Bui, 2010), could rapidly become important components of the core of ant-plant networks even if they do not displace other ant species (Díaz-Castelazo et al., 2010; Falcão et al., 2017).

While the importance of abiotic/biotic factors have been separately explored for ant-plant networks, the simultaneous relative contribution of biological attributes of species and ecological and habitat level attributes (i.e., ecological correlates) in a facultative mutualistic ant-plant network, is addressed here for the first time. Attributes of the species sets considered here are in accordance with the foraging theory perspective required for a mechanistic understanding of ecological networks (Ings et al., 2009). Our study system provides the opportunity to test simultaneously the effect of several ecological and biological attributes of interacting species, including morphology, behavior, and abundance as well as their inter-habitat (spatial) variation, on the overall and ‘fine-grain’ structure of a quantitative mutualistic network. In particular we addressed the following questions: (1) What is the network structure of this intensively-sampled ant-plant community-mediated by extrafloral nectaries? (2) Which is the “fine-grain” structure emerging from the frequency (strength) of pairwise interactions? (3) Which is the position of species in the core/periphery structure of the network? (4) Which is the relative contribution of biological or ecological correlates (ant, plant, or habitat attributes) in rendering the “fine-grain” and overall network structure?

Materials & Methods

Study site and data collection

Field work was carried out in the Centro de Investigaciones Costeras La Mancha (CICOLMA), located on the coast of the state of Veracruz, Mexico (19°36′N, 96°22′W; elevation <100 m). The climate is warm and sub-humid with rainy season between June and September, a total annual precipitation is ca. 1,500 mm, and mean annual temperature is 22°−26 °C (Rico-Gray, 1993). The major vegetation types in the study area are tropical sub-deciduous forest, tropical deciduous forest, coastal dune scrub, mangrove forest, freshwater marsh, and deciduous flood forest (Castillo-Campos & Travieso-Bello, 2006). Changes in the abundance of associations between ants and plants bearing EFNs suggest that ant–plant interactions are strongly influenced by climatic conditions as a result of marked seasonality (Díaz-Castelazo et al., 2004; Rico-Gray & Oliveira, 2007). Marked seasonality at the study site (rainy, dry, and cold-front seasons) influences primary productivity and have a strong effect in animal-plant interactions (Rico-Gray, 1993; Díaz-Castelazo et al., 2004; Sánchez-Galván, Díaz-Castelazo & Rico-Gray, 2012; López-Carretero et al., 2014; Ramos-Robles, Andresen & Díaz-Castelazo, 2016; Martínez-Adriano, 2017).

Biweekly observations were conducted between October 1998 and September 2000 (Rico-Gray, 1993; Díaz-Castelazo et al., 2004), rendering an intensive sampling of 48 censuses along six 1 km trails that sampled vegetation types representative of the plant communities in the study area: (1) pioneer dune vegetation (PDV), (2) coastal dune scrub (CDS), (3) tropical sub-deciduous forest in young soil (TSF-Y), (4) tropical sub-deciduous forest in old soil (TSF-O), (5) tropical deciduous flood forest with wetland (TDF-W), and (6) mangrove forest ecotone (MFE) (nomenclature as in Martínez-Adriano, Aguirre-Jaimes & Díaz-Castelazo, 2016; derived from Castillo-Campos & Travieso-Bello, 2006, following methods from Rico-Gray, 1993; Díaz-Castelazo et al., 2004). Vegetation associations differ in their structural complexity provided partly by arboreal plant cover and contrasting physiognomies occur between “open” and “shaded” habitats. In CICOLMA the first three habitats (PDV, CDS, and (TSF-Y) being included in the former physiognomy and the other three habitats (TSF-O, TDF-W, and MFE) included in the latter (Díaz-Castelazo et al., 2004; López-Carretero et al., 2014). Habitats 1, 2, and 3 are also different from 4, 5, and 6 in their floristic similarity of flowering plants (Chao-Jaccard Similarity Index, see Martínez-Adriano, Aguirre-Jaimes & Díaz-Castelazo, 2016) and in the mean density of ants observed in honey baits placed in Díaz-Castelazo et al. (2004). In these six vegetation types we recorded all occurrences of ants collecting liquids directly from all plants within each transect (ant-plant interactions). We considered all plant life forms but only from those below 4 m in height, since no canopy censuses were performed.

We also estimated the abundance of EFN-bearing plants through their line cover within each transect (please see Díaz-Castelazo et al., 2004 or Sánchez-Galván, Díaz-Castelazo & Rico-Gray, 2012 for details). On each visit at each transect we recorded: ant species, plant species, the plant organ where the extrafloral nectaries were located, and its distribution. Once an individual plant was marked as visited by ants, it was subsequently re-checked throughout the study. When doubt existed on the nectar source, EFN-secretion, we corroborated this with glucose reagent strips (Clinistix, Bayer). We considered extrafloral nectar either produced by the surface of reproductive structures such as the spike, pedicel, bud, calyx, or fruit, or secreted by special structures on vegetative parts such as leaves, shoots, petioles, bracts, or stems. Ants were considered to be feeding on nectar when they were immobile, with mouthparts in contact with nectar secreting tissues for periods of up to five minutes (Rico-Gray, 1993), although for some species, particularly when recruitment of nestmates to the nectar source occurred, ant feeding was very evident and thus, recorded in shorter time periods. Further information on the ant-plant interaction censuses showed at the present study (including seasonal variations of species and attributes) is detailed in Díaz-Castelazo et al. (2004) (Appendix S1, Fig. 1).

Figure 1 Quantitative mutualistic networks between EFN-bearing plants (lower trophic level, green nodes) and ant visitor species (higher trophic level, red nodes).

Blue-colored nodes depict species constituting the core of the network. Species codes as in Tables 1 and 2.

Table 1 EFN-bearing plant species within the network and its attributes.

Plant attributes considered also in Fig. 2 are: EFN, Distribution of extrafloral nectaries within a plant species (‘C’ are circumscribed glands and ‘D’ are disperse glands). Habitat, distribution of plant species between habitats with contrasting vegetation structure (‘S’ is shaded vegetation and ‘O’ is open vegetation). Abundance, percentage cover of EFN-bearing plant species.

Plant species	Plant species code	Distribution of EFNs	Habitat structure	Abundance (%cover)	
Cordia spinescens	CorSpi	D	S	38.833	
Turnera ulmifolia	TurUlm	C	O	6.66	
Crotalaria indica	CrotIn	C	O	12.38	
Cedrela odorata	CedOdo	D	S	36.143	
Callicarpa acuminata	CallAc	D	B	68.797	
Caesalpinia crista	CaeCri	C	O	27.15	
Bidens pilosa	BidPil	C	S	27.95	
Canavalia rosea	CanRos	C	O	76.057	
Calopogonium caeruleum	CalCae	C	O	16.85	
Terminalia catappa	TerCat	C	S	0.35	
Senna occidentalis	SenOcc	C	S	3.717	
Opuntia stricta	OpuStri	D	O	64.35	
Hibiscus tiliaceus	HibTill	C	O	2.4	
Amphilophium paniculatum	AmphPa	D	O	17.55	
Ipomoea pescaprae	IpoPes	C	O	49.1	
Conocarpus erectus	ConEre	C	S	16.383	
Ficus obtusifolia	FicObt	C	S	8.15	
Cornutia grandiflora	CorGra	D	O	2.5	
Macroptilium atropurpureum	MacAtr	C	O	16.3	
Cissus rhombifolia	CisRho	C	O	3.55	
Ipomoea sp.	IpoSp.	C	S	12.167	
Mansoa hymenaea	ManHym	C	S	16.3	
Tabebuia rosea	TabRos	D	S	6.66	
Acacia macracantha	AcaMac	C	B	2.75	
Trichilia havanensis	TriHav	C	S	28.33	
Arundo donax	AruDon	C	O	151.66	
Petrea volubilis	PetVol	D	O	74.1	
Chamaecrista chamaecristoides	ChaCha	C	O	32.4	
Iresine celosia	IreCel	C	O	16.55	
Cordia dentata	CorDen	D	S	3.615	
Bunchosia lindeliana	BunLin	C	S	1.7	

Table 2 Ant species within the network and its attributes.

Ant attributes considered also in Fig. 2 are: invasive status, status as invasive/tramp ant species (INV or NO). Dominance, hierarchies of behavioral dominance (from the most dominant to the least) are: A, Dominant Dolichoderine; B, Generalized Myrmicine; C, Subordinate Camponotini; D, Tropical Climate Specialists; E, Opportunistic; F, Specialist Predators; and Head Length, length (mm) from head apex to anterior clypeal margin of species (minor worker).

Ant species	Ant species code	Invasive status	Dominance hierarchy	Head length	
Camponotus planatus	CamPla	NO	C	1.198	
Camponotus mucronatus	CamMu	NO	C	1.418	
Camponotus atriceps	CamAt	NO	C	1.946	
Azteca sp. 1	AztSp	NO	A	1.471	
Paratrechina longicornis	ParLo	INV	E	0.638	
Tetramorium spinosum	TetSpi	INV	E	0.968	
Cephalotes minutus	CepMin	NO	D	1.155	
Dorymyrmex bicolor	DorBi	NO	A	0.973	
Pseudomyrmex gracilis	PseGra	NO	F	1.738	
Monomorium cyaneum	MonCy	NO	B	0.482	
Camponotus mucronatus hirsutinasus	CamHi	NO	C	1.076	
Pachycondyla villosa	PachVi	NO	F	2.880	
Forelius analis	ForAna	NO	A	0.631	
Crematogaster brevispinosa	CreBre	NO	B	1.031	
Pheidole sp.	PheSp	NO	B	0.553	
Solenopsis geminata	SolGe	INV	D	0.684	
Wassmannia auropunctata	WasAu	INV	D	0.479	
Pseudomyrmex ejectus	PseEje	NO	F	0.800	
Pseudomyrmex brunneus	PseBru	NO	F	0.768	

Plant and ant attributes

Regarding the distribution of EFNs among plant organs, we used a general characterization (see Díaz-Castelazo et al., 2005) differentiating the EFNs which are glands circumscribed to particular plant organs or whorls (at specific or modular locations) from the ones dispersed among plant organs (i.e., secretory trichomes on leaves or surfaces of vegetative tissues). Díaz-Castelazo et al. (2005) results raised the idea that gland distribution on plant organs could follow an aggregate (i.e., circumscribed) location against a widely dispersed location and this could result in distinct ant visitor arrays (Díaz-Castelazo et al., 2004). In a similar way that extrafloral nectar sources may differ from honeydew sources in their associated ant assemblages (Blüthgen & Fiedler, 2004; Blüthgen et al., 2000). Circumscribed EFNs include: elevated glands, hollow glands (vascularized), transformed glands (vascularized), capitated trichomes (non-vascularized), and unicellular trichomes (non-vascularized). Disperse EFNs include: flattened glands, peltate trichomes, and scale-like trichomes (Díaz-Castelazo et al., 2005).

Figure 2 Ordination of NMDS representing the assemblage of interactions given the ant-plant distances (Bray–Curtis) at the network.

Plant species in black; ant species in red. Species names appear as in Tables 1 and 2: at the ordination, first plant species (P1), second plant species (P2) and so on, correspond to the first plant species, and the second plant species in Table 1 and so on. First ant species (A1), second ant species (A2) and so on, correspond to the first ant species, and the second ant species in Table 2 and so on. NMDS Stress = 0.17 (fourth iteration) indicates a good two-dimensional solution of the ordination suitably representing ant-plant assemblage dissimilarity; this configuration also has very low residuals (max res = 0.0004) showing a good concordance between the calculated dissimilarities and the distances among objects. Non-overlapping ellipses (i.e. orange and green) circle the attribute (factor) that significantly explained (r2 = 0.24, P = 0.005) the pairwise interaction pattern (habitat types).

Attributes for plants included: (1) the abundance of plants with EFNs, (2) species distribution in vegetation associations with distinct habitat structure (open or shaded habitats), and (3) the distribution of the EFNs among plant organs (Table 1). Attributes for ants included the following: (1) behavioral dominance based in the classification of ant functional groups proposed by Andersen (1995) and Andersen (2000) in relation to plant life-forms, stress, and disturbance, (2) head length, a robust estimator of body mass in ant species (Kaspari & Weiser, 1999), and (3) species status as invasive. The invasive status that we used was based in Holway et al. (2002), with adjustments to include ‘tramp’ species status as well. Invasive ants are those non-native ant species which establish long-term populations and expand their range upon introduction to new areas, while tramp ants are non-native transferred populations of ants closely tied with urban areas and human activities (considered thus as “human commensals”) (McGlynn, 1999; Lach & Hooper-Bui, 2010; Falcão et al., 2017); (Table 2).

We provide further detail on species attribute selection at the present study as follow. Cover and distribution of EFN-bearing plant species (among habitats with different vegetation structures) is an important factor influencing the richness and abundance of ant-plant interactions (Díaz-Castelazo et al., 2004), interactions with other insects (López-Carretero et al., 2014), and the spatio-temporal variations due to seasonality (Rico-Gray, 1993; Rico-Gray & Oliveira, 2007; Díaz-Castelazo et al., 2010). Similarly, the differential distribution of EFNs among plant organs could favor different ant assemblages (Majer, 1993; Blüthgen & Fiedler, 2004). This is essential for the optimal defense of valuable reproductive plant organs compared to vegetative ones (Rico-Gray, 1993; Wäckers & Bonifay, 2004; Holland, Chamberlain & Horn, 2009). Related to ant attributes, behavioral dominance is a relevant feature in mutualistic ant-plant interactions given that competitive species may exclude submissive ones (Andersen, 2000; Ness & Bronstein, 2004; Lach & Hooper-Bui, 2010; Dáttilo, Díaz-Castelazo & Rico-Gray, 2014c). For example, head length has been shown to be positively correlated with the number of plant species that each ant species interact in ant-plant networks (knows as “degree”; Chamberlain & Holland, 2009). Other relevant attribute that could affect the ant-plant interactions is the presence of invasive ant species since many invasive species have behavior or foraging strategies that overcome their native counterparts (Díaz-Castelazo et al., 2010) or disrupt the mutualistic interactions (Schultz & McGlynn, 2000; Holway et al., 2002).

Data analysis

The ant-plant network analyzed here consists of a quantitative species-species matrix given by the frequency of occurrence of each pairwise ant-plant interaction. Ecological and biological attributes of the species were of different kinds and considered as highly important in modulating the mutualistic interaction (Díaz-Castelazo et al., 2004; Díaz-Castelazo et al., 2005).

The pairwise interaction matrix here considered is a highly informative sub-web taken from Díaz-Castelazo et al. (2010), where we excluded those interactions that occurred at considerably low frequencies (interactions recorded on less than three occasions from the whole 48 censuses), in order to perform better multivariate analysis (NMDS), interpretation of biplot ordinations, and adjustment of explanatory variables. This also reduced the probability of considering a species with a single or very few interactions as a “specialist”, when it was just a very rare species and helped to avoid the overestimation of specialization, nestedness, and strength asymmetry (Blüthgen et al., 2008).

For this informative network we analyzed nestedness (NODF) (Nestedness based on Overlap and Decreasing Fill) (Almeida-Neto et al., 2008) using ANINHADO (Guimarães & Guimarães, 2006). This metric is robust to detect a nested pattern since it is less sensitive to matrix size and shape than other measures such as nestedness derived from matrix Temperature (Almeida-Neto et al., 2008). Significance of the NODF value for our network was obtained with ANINHADO after comparing it with 1,000 simulations using null model Ce (Guimarães & Guimarães, 2006), which corresponds to Null Model II of Bascompte et al. (2003). It assumes that the probability of occurrence of an interaction is proportional to the observed number of interactions of both plant and ant species (Bascompte et al., 2003; Dáttilo, Guimarães & Izzo, 2013b). We then estimated network topology or structural metrics (connectance, dependence asymmetry, weighted nestedness, and niche overlap) using different indexes included in the function “network-level” of the “bipartite” package (Dormann & Gruber, 2009) in ‘R’ software ver. 3.5.1 (R Core Team, 2014; Dormann et al., 2009).

Additionally, with the software MODULAR (Marquitti et al., 2014) we tested the existence of a modular structure in the network with the modularity index (M). This index ranges 0–1 and was calculated with simulated annealing optimization approach (SA) (Guimerà & Amaral, 2005). This metric was based on Barbers modularity metric which are recommended for bipartite networks (QB)(Barber, 2007). The statistical significance of modularity (M) was calculated using Monte Carlo tests with 1,000 randomizations (Guimerà, Sales-Pardo & Amaral, 2004). High values of M indicate the occurrence of ants and plants in cohesive subgroups that generate compartments or modules in which these species interact more closely than with the other species in the network (Olesen et al., 2007).

For calculation of the “fine-grain” structure of the network we used the “species-level” function (Dormann, 2011) in the “bipartite” package. The metrics calculated for this objective were “species strength” and d’. The first is defined as the sum of dependences of the plants visited by this animal (or vice versa). Thus, species strength is a quantitative extension of the metric “species degree” and provides information about the relevance of a species for their interacting counterpart, being thus a meaningful measure of network complexity (Bascompte, Jordano & Olesen, 2006). While, d’ is the specialization of each species based on its discrimination from random selection of partners (Blüthgen et al., 2008). Finally, we calculated core–periphery structure of the network and its component species (i.e., which species constitute the cohesive core are generalists, and which the low-degree species constitute the peripheral). This metric was calculated with a function developed by Martínez-Adriano (2017) in R software based on the formula proposed by Dáttilo, Guimarães & Izzo (2013b), where the species with values equal or larger than 1 are considered as core components and species <1 are considered peripheral.

In order to explore the among-species dissimilarities resulting in the interaction pattern of the network, we generated the ordination of interaction frequencies with “Non-metric Multidimensional Scaling (NMDS)” multivariate technique (Quinn & Keough, 2002). This method is specifically designed to graphically represent relationship between objects (i.e., species/sites) in a multidimensional space provided by non-metric dissimilarities among objects (Quinn & Keough, 2002). NMDS is one of the most effective methods for the ordination of ecological data and the identification of underlying gradients because it does not assume a linear relationship among all variables (Quinn & Keough, 2002). NMDS reduces the dimensionality of a matrix among sample similarity coefficients, based on particular number of dimensions (Borg & Groenen, 1997). We chose the Bray–Curtis dissimilarity coefficient to construct the similarity matrices because joint absences do not influence among sample similarity, and then we chose a two-dimension configuration. The fit of an NMDS ordination, known as “stress”, is determined by how well the ordination preserved the actual sample dissimilarities, where values range from zero to one (values of 0.2 and below are valid configurations to be interpreted). Because NMDS analysis offers more than one solution, we carried out an iterative process to find the model with smallest stress value using the metaMDS function in “Vegan” package (Dixon, 2009) for R software (R Core Team, 2014).

In order to explore the simultaneous relative contribution of several biological and ecological species attributes and habitat level attributes on the interaction pattern (NMDS ordination), we fitted those ecological/biological factors and vectors using the envfit function from the “Vegan” package (Dixon, 2009) on R software (R Core Team, 2014). This function fit the vectors (continuous variables) and factors (categorical variables) from the environmental variables to the NMDS ordination, providing statistical significance by comparing our real model of pairwise interactions with 1,000 permutations of a given null model. The envfit function provides a measure of correlation (r) and a significance value (p) based on the probability that 1,000 random permutations of simulated (environmental) variables would have a better fit than the real variables (Oksanen, 2009).

To test if the frequency of ants was different when foraging in the different EFN morphological types, we performed a χ2 test. To test if between-group floristic similarity (Sorensen’s floristic similarity index, Češka, 1966) was higher than within group floristic similarity we performed one-way ANOVA contrasting open and shaded habitats. With this analysis we further confirm that open and shaded habitats differ in their vegetation structure and in turn, provide differential biotic and abiotic conditions for inhabitant species, presumably affecting the resultant network structure of ant-plant interactions. To explore if there was a relation between ant head length and species degree in the network (the number of plant species interacting with ants), we performed a Spearman rank correlation test (Quinn & Keough, 2002).

Results

Network-level and fine-grain structure

Our ant-plant network involved 31 EFN-bearing plant species and 19 ant-forager species linked by 1,302 quantitative interactions (overall frequency of interactions) derived from 157 species associations (links among species). The general topology shows a highly and significantly nested network (NODFObs = 49.13, NODF(Ce) = 34.93, P(Ce) < 0.001). Although five modules were detected in the modularity analysis (Barber’s QB) the network was not significantly modular (M = 0.288, P = 0.55), thus no true compartments exist in the network (Fig. 1). Network-level indexes were: connectance = 0.267, dependence asymmetry = 0.669 (implying that plants depend more on ants than the opposite), niche overlap among ant species = 0.223, niche overlap among plant species = 0.425, and weighted nestedness = 0.554 (implying that the network is still nested when considering the frequency of pair-wise interactions). Four plant species and three ant species constituted the central core of this network, the remaining species were peripheral. Plant core species were: Cordia spinescens, Cedrela odorata, Callicarpa acuminata, and Crotalaria incana, while ant core species were: Camponotus planatus, Camponotus mucronatus, and Camponotus atriceps.

In terms of ‘species strength’, most plant species exhibited low strength values (below 1), thus having a modest relevance for the ant community. However, some plant species stand out with higher strength values (around 2) which are Cordia spinescens, Cedrela odorata, Callicarpa acuminata, and Turnera ulmifolia. These plant species are the most important EF nectar sources for ant foragers at a community level. Species-level specialization values (d’, considered as a measure of selectiveness) for plant species were also generally low (around 0.1), and only those plant species with few associated ant species (ant species with interaction patterns atypical or different from the rest) showed values above 0.3, such as Ficus obtusifolia (d′ = 0.43) and Senna occidentalis (d′ = 0.37). In contrast with plants, some ant species had higher strength values. Seven ant species had values above 1, and two core ant species, Camponotus planatus and C. mucronatus, have strength values over 6, being thus very important visitors of EFN-bearing plants. These ant species with high-strength values are relevant at a community level. On the other hand, specialization (selectiveness) for ants was generally low (around 0.3, thus, higher than plants), but few ant species had intermediate d’values such as Tetramorium spinosum (d′ = 0.53) or Wasmannia auropunctata (d′ = 0.43) not only because they interact with few plant species but with plants visited by few ant species.

Relative contribution of attributes to the assemblage of pair-wise interactions

Attributes of species are summarized as follows: plant species with circumscribed nectaries (Table 1) produced larger mean nectar volumes (2.06 µl), than those plants with dispersed nectaries (0.53 µl). However, the number of active glands in a plant individual may be higher for dispersed nectaries, since these glands are structurally simpler than those of circumscribed nectaries. The frequency of ants foraging on the different EFN morphological types (Díaz-Castelazo et al., 2005; Díaz-Castelazo, Chavarro-Rodríguez & Rico-Gray, 2017) were different (χ28 = 1,091.7, P <  0.01). Moreover, the range of total associated ant species visiting plants considered to have each type of nectary is different among EFN distribution types. The range of visits to circumscribed nectaries was between nine and 17 ant species, while it was between 20 and 23 ant species for disperse nectaries.

We considered the two main vegetation structural associations (“open” vs. “shaded habitats”) to be natural groups (Table 1), because floristic similarity between them is considerably lower and significantly different (F1,13 = 15.79, P < 0.01) to that occurring within each group (36.06 and 41.28 for open and shaded habitats, respectively). See Methods for information on the vegetation associations, either of “open or shaded habitats”.

The stress value of the multivariate Non-Metric Multidimensional Scaling (NMDS) obtained at the fourth run of the iterative process was the lowest (0.17), and suggests that the NMDS two-dimensional solution of the ordination suitably represented ant-plant assemblage dissimilarity. This configuration also had very low residuals (max res = 0.0004) indicating a good concordance between the calculated dissimilarities and the distances among objects. In Fig. 2, axis NMDS1 is related to the contribution or importance of plant species to the ant forager community. Those plant species that are ordered at the extremes of axis NMDS1, either with low (negative) or higher (positive) values for component NMDS1, have low species strength values; thus, these plant species have ‘atypical’ interaction patterns (and occupy in Fig. 1, the lowest or basal position within the network). In contrast, at axis 1 of NMDS, those plant species aggregated near zero are those plant species with the highest species strength or relevance for the associated ant community. For ants, no generic or grouping trends are apparent. Axis NMDS2 divides plant species according to the main habitats where they occur. Starting from the upper part of the ordination and ending at the lower part, we see an arrangement of the plant species according to a ‘humidity’ gradient of habitats. First we see plants of shaded habitats with modest light incidence and higher humidity and at the lower part of the biplot along axis 2 we see the plants of the drier habitats. The higher values for NMDS2 show (in decreasing order) plants (and associated ants) from the MFE, followed TSF-W, and TSF-O. At the bottom of the bi-plot, the plants and ants occurring mostly in open vegetation types with high light incidence: from zero to the lowest values of NMDS2, the interacting species are arranged through TSF-Y, CDS, and PDV.

The results of fitting among the biological/ecological variables and the NMDS ordination showed that vegetation associations with differential structure (open vs. shaded habitats) were the variables that determined the variation in the frequency of ant-plant pairwise interactions-mediated by EFNs (r 2 = 0.24, P = 0.005). Two contrasting groups were formed along NMDS2, which were plant species (and their associated ant forager species) located either in open or shaded habitats (Fig. 2). Neither the distribution of EFNs on plant organs, nor the abundance of extrafloral-nectary bearing plants at each vegetation type, had a significant contribution to the variation in the observed ant-plant association patterns.

None of the variables of ant species (behavioral dominance, invasive status or head length) showed in Table 2, explain the network’s fine-grained structure. However, we found a significant positive correlation between ant head length and species degree (the number of plant species interacting with ants) (Spearman rank correlation, rs = 0.565, P < 0.05). A trend at the NMDS ordination is that the invasive (tramp) ant species in our study (Solenopsis geminata, Wasmannia auropunctata, Tetramorium spinosum, and Paratrechina longicornis) separate from the rest of ant species at the interaction display. However, when all ant attributes are simultaneously considered, they do not provide significant contribution to the variation in the observed ant-plant association pattern.

Discussion

Network-level and fine-grain network structure

Our studied network, comprising 31 plant and 19 ant species attached by 157 interaction links, has a general nested structure and is thus asymmetric in its specialization patterns (see also Díaz-Castelazo et al., 2010; Díaz-Castelazo et al., 2013). The network shows no modular structure, as the non-modular structure that occurs commonly in theoretical mutualistic networks, especially for facultative non-symbiotic interactions (Guimarães et al., 2007). Few species with very high interaction frequencies exists within our network (eight plant and four ant species), referred as ‘super-generalists’. Super-generalist species are fundamental components for the maintenance of convergence at the community level within highly diversified mutualistic assemblages (Guimarães, Jordano & Thompson, 2011). In our study, super-generalist species may favor trait convergence. That is, core ant species belong to the same functional group (Subordinate Camponotini) and share adaptations for foraging on plant-derived liquids resources such as extrafloral nectar (Davidson, Cook & Snelling, 2004). Similarly, core plants species show mostly “disperse” EFNs, a gland distribution that may favor a more diverse array of associated ant visitors.

In our study system, the fact that the plant ‘guild’ shows higher dependence asymmetry values than ants, implying that the studied community plants ‘depend’ more on ants as a guild than the opposite. It is also reinforced by the higher species strength values of ants than those given for plants. This asymmetry could reflect a higher temporal turnover of plants at the network —perhaps caused by seasonality or disturbance versus higher ant resilience—probably derived from facultative foraging of ants. Three of four plant species constituting the core of this network had high strength values (Cordia spinescens, Cedrela odorata, and Callicarpa acuminata); these results suggest that the most connected plant species are important resources for the ants at the community level. However, the relative importance of specific plant species for this ant community do not seem related to specific biological attributes or neutral effects since neither mean nectar volumes secreted by each plant species (Díaz-Castelazo et al., 2005; Díaz-Castelazo, Chavarro-Rodríguez & Rico-Gray, 2017), nor gland distribution or plant abundance explained core composition and species strength of plant species. Instead, this pattern seems to emerge from degree and interaction frequencies, possibly driven by other higher-scale factors (i.e., habitat structure, species co-occurrence, abiotic variables, etc.).

The rest of plant species showed very low species strength values, having thus a modest relevance for the ant community. Species-level specialization values (d′) for plant species were also generally low (around 0.1) and only those plant species with few associated ant species (with an atypical interaction pattern) exhibited values above 0.3. These findings are in accordance with the generalized, highly nested structure of this network. For potentially mutualistic networks (such as this) and for facultative ant-plant interactions (such as those mediated by extrafloral nectar), low specialization or selectiveness for each species (and the whole network) is the general trend (Bascompte et al., 2003; Vázquez & Aizen, 2004; Díaz-Castelazo et al., 2010).

In contrast with plants (which have lowest species strength values), the ant species that constitute the core of this network (Camponotus planatus, Camponotus mucronatus, and Camponotus atriceps) had species strength values above 1. Camponotus planatus and C. mucronatus have strength values above 6, being thus remarkably important visitors of EFN-bearing plants. Species belonging to this genus are frequent visitors of EFNs (Díaz-Castelazo et al., 2004; Díaz-Castelazo et al., 2013) and solitary leaf foragers that cover high foliar areas. Camponotus species have high ability to rapidly take up nectar given their proventricular adaptations that allow passive damming of sugary liquids, large crop capacities, and seeping canals to nourish the midgut (Davidson, Cook & Snelling, 2004). Thus, this group of ants is highly adapted to forage on nectar and sugary liquids. It is understandable that at the present study Camponotus species have high degree, high strength values, and low levels of specialization or selectiveness (d′). Given that these ants are physiologically adapted to forage in all available extrafloral nectar sources and not having any trophic restriction, they tend to be generalist visitors of EFN-bearing plants. Although some other ecological aspects—such as competition ability of other ant species and resource attractiveness (Dáttilo, Díaz-Castelazo & Rico-Gray, 2014c)—may differentiate visitation pattern of these core ant species. In our studied community core ant species are relevant to most plants species at the network and the plant species depend more of ant species than the opposite. Adaptations to nectar consumption and deterrent behavior of core ants (as well as their high interaction frequency), suggest that these species may provide potential biotic defense at a community scale; this do not exclude the possibility that many other ant species provide biotic defense at smaller scales or for particular plant species.

Relative contributions of the attributes to the assemblage of pairwise interactions

Major vegetation associations grouped according to habitat structure were the only factors that explained variations in pair-wise interactions or fine-grain structure of the network. Open and shaded habitats at the study site seem to differ structurally in vegetation and on their abiotic conditions, which may in turn be important determinants for insect-plant interactions (López-Carretero et al., 2014). Although, some studies have discussed the possible effects of abiotic variables on ant-plant networks (Díaz-Castelazo et al., 2010; Rico-Gray & Oliveira, 2007) (references therein; Rico-Gray et al., 2012; Sánchez-Galván, Díaz-Castelazo & Rico-Gray, 2012), our study is the only one addressing habitat abiotic effects jointly with species-level biological attributes and neutral explanations (i.e., abundance) in a quantitative ant-plant network.

A mechanistic explanation for the differential ant-plant association pattern between open and shaded habitats (suggesting habitat complexity effects; Dáttilo, Guimarães & Izzo, 2013b), could include light incidence (under light conditions, jasmonic acid-induced EFN secretion is higher than in dark conditions), ‘attractiveness’ or nutritional value of extrafloral nectar secreted by ‘light demanding’ plant species compared to ‘shade tolerant’ ones, and the physiological tolerance of ants to high temperatures (Radhika et al., 2010). Increased photosynthetic activity of plants in open light-rich habitats could result in higher carbohydrate availability in extrafloral nectar and thus increased attractiveness to ants (Radhika et al., 2010), or a higher density of EFN-bearing plant life forms (such as vines). EFN-bearing plants growing in sunlight obtain a measurable benefit from ant visitation, whereas the same plant species growing under shaded conditions has no such a benefit (e.g., Bentley, 1976; Frank & Fonseca, 2005). For some plant species size of EFNs and nectar secretion are higher under intense light conditions compared to low light conditions (Yamawo & Hada, 2010) and a similar trend is found for the ant abundance foraging on these glands (Rudgers & Gardener, 2004; Yamawo & Hada, 2010). This effect of site conditions on EFN abundance and secretory activity could also exist in our study system since vegetation types with canopy cover (shaded) versus open habitats do sustain different species abundances, floristic similarities (Díaz-Castelazo et al., 2004), and patterns of specific insect-plant interactions (López-Carretero et al., 2014).

Among-habitat heterogeneity in vegetation structure (as well as seasonality) in our study site is so prevalent (having a strong effect in animal-plant interactions as seen in Rico-Gray, 1993; Díaz-Castelazo et al., 2004; Sánchez-Galván, Díaz-Castelazo & Rico-Gray, 2012; López-Carretero et al., 2014; Ramos-Robles, Andresen & Díaz-Castelazo, 2016). This is clearly detected in the ant-plant interaction pattern, in contrast to other studies where vegetation structure differences are not so outstanding as have an effect in other ant-plant networks (Dáttilo, Guimarães & Izzo, 2013b). Further evidence of among-habitat heterogeneity translating to ant-plant network structure is provided in the present study by the multivariate analysis. In this analysis one of the components explaining the variance in the lack of independence among ant and plant species (NMDS2) displays habitats following a decreasing humidity gradient, from MFE, followed by TSF-W and TSF-O, TSF-Y, CDS and PDV. Indeed, open habitats at the study site, such as coastal dune scrub (CDS) and pioneer dune vegetation (PDV) have the most extreme temperatures, solar radiation (Moreno-Casasola, 1982; Moreno-Casasola & Travieso-Bello, 2006), and are exposed to continual disturbance (López-Carretero et al., 2014) like sand movement, strong winds, and abrasion (Pérez-Maqueo, 1995).

Our results showed that no neutral effects derived from variation in species abundances are structuring the studied ant-plant network. Abundance of EFN-bearing plant species was considered in our analysis but rendered no significant contribution to explain the frequency of pairwise ant-plant interactions. Similar results were found for another ecological network at the same study site such as a plant-herbivore network (López-Carretero, et al., 2016), where network parameters were not influenced by plant cover (abundance) but by biological and seasonality aspects. In our study, although ant abundance was not included, we know from robust estimates of ant density (honey baits) at the same periods of time and vegetation types that average ant density is higher in open habitats than in shaded ones (Díaz-Castelazo et al., 2004). In other studies of ant-plant interactions, the abundance of interacting species partially explain some features of network structure (Vázquez et al., 2007; Dáttilo et al., 2014a). However, Dáttilo et al. (2014a) show that although more abundant ant species interact with more plant species with EFNs, information on the difference in abundance among interacting species was insufficient to explain ant-plant network organization. That is, nestedness was higher in networks of ants and plants with EFNs than that observed in networks of ants and plants without EFNs. Thus, the differences in nestedness, connectance, and heterogeneity of interactions remained after controlling for the effects of species richness structure.

Other potentially mutualistic networks have shown that species abundance or temporal overlap is far from accurately predicting the frequency of pair-wise interactions (Jordano, Bascompte & Olesen, 2006; Vázquez et al., 2009). Poisot, Stouffer & Gravel (2015) outline several direct (abundance-based and trait-based) and indirect (biotic modifiers and indirect effects of co-occurrence) effects to account for variation in interactions occurrence. Given that perspective, at the sampling intensity and duration of our ant-plant interaction survey (this reflect temporal and spatial variation; see Díaz-Castelazo et al., 2004), neither the abundance-based nor the trait-based modifiers seems to be enough relevant to explain the variation in pairwise ant-plant interactions, even if at other scales ant abundance could partially explain an overall network pattern (Dáttilo et al., 2014a). In contrast, an indirect effect given by habitat structure (biotic modifiers through co-occurrence, sensu Poisot, Stouffer & Gravel, 2015) more thoroughly explains the quantitative interaction pattern at the present study.

The fact that neither the distribution of EFNs on plant organs nor the abundance of extrafloral-nectary bearing plants at each vegetation type had a significant contribution to the variation in the observed ant-plant association patterns, does not rule out its potential effect on ant foraging patterns in other ecosystems or spatial scales (Dáttilo et al., 2013a; Dáttilo et al., 2014b). At our study site, besides the overwhelming evidence of seasonality and habitat heterogeneity (we did find an effect of habitat structure in ant-plant interactions within the network), the high occurrence frequency and seasonal activity of morphologically diverse EFNs at vegetation associations (Díaz-Castelazo et al., 2004; Díaz-Castelazo et al., 2005) suggests a temporal variation in benefits provided by ant visitors to EFN-bearing plants. Indeed, at the studied habitats, several plant species receive anti-herbivory defense from ants foraging on EFNs (Oliveira et al., 1999; Cuautle & Rico-Gray, 2003; Chavarro-Rodríguez, Díaz-Castelazo & Rico-Gray, 2013). There is also evidence that the frequency of ants foraging on different EFN morphologies and distributions among plant organs differs (Díaz-Castelazo, Chavarro-Rodríguez & Rico-Gray, 2017). Other anti-herbivory plant defenses of plant species at the study site have shown spatial and temporal variation (López-Carretero, et al., 2016; López-Carretero et al., 2018). Thus, information on the contribution to plant fitness of the EFN occurring among plant organs (and their temporal activity patterns) could shed light on the optimal defense-value of EFN resources as an indirect defense (Holland, Chamberlain & Horn, 2009) mediated by ants, an issue not yet explored at a community-level.

For ant variables, although ant size (head length) was not a significant factor explaining frequency of pairwise interactions, it was important in explaining other attributes such as species degree within our mutualistic network. This may occur because competition among ant species foraging at EFNs could vary with ant body size, size of ants contributing thus to the species degree values (Chamberlain & Holland, 2009). Larger ant species can forage over a greater area than small species, and thus interact with more plant species. In addition, it has been shown that recruitment of ant foragers to a resource is negatively correlated with ant body size (LeBrun, 2005). That is, while ant body size increases, the number of recruiting foragers decreases, which can lead to a body size-driven competition hierarchy in which larger ant species visit more plant species.

Overall, behavioral dominance as a factor was not significant to explain variations in the frequency of pair-wise interactions, possibly due to the spectrum of factors considered simultaneously within the analysis since the EFNs considered here include both disperse EFNs and circumscribed EFNs that could provide resources for both dominant and non-competitive ant species. However, ant invasive/tramp species in the study site (Solenopsis geminata, Wasmannia auropunctata, Tetramorium spinosum, and Paratrechina longicornis) seem to have a slightly different pattern of interaction from the other species (separate from the rest of ant species in the interaction display), probably due to their ability to access new habitats or food resources (Ness & Bronstein, 2004; Lach & Hooper-Bui, 2010). This makes sense in such a human-altered ecosystem as La Mancha, that seems to rapidly reflect ant invasions. At smaller time-scales, in the same study site ant invasiveness does not alter the core structure of the network (Falcão et al., 2017), despite of other possible functional effects in the community that are just about to be explored.

Conclusions

Our extrafloral-nectary mediated ant-plant network was highly nested, non-modular, showed high species strength for core species, low specialization or selectiveness, and higher dependence of plants on ants. These results are in accordance to a facultative mutualism scenario, mainly considering that the core ant species in this interaction network are known as good plant-defenders in general (Oliveira et al., 1999; Cuautle & Rico-Gray, 2003; Dáttilo, Díaz-Castelazo & Rico-Gray, 2014c).

When simultaneously exploring plants, ants, and habitat attributes on a network-level and fine-grain structure, the only factor that significantly affects the pair-wise interactions is habitat heterogeneity in the vegetation structure (and distribution of EFN-bearing plant species). At our study site this heterogeneity is so strong that is clearly detected in the ant-plant interaction network patterns, both in network topology and in the fine-grain network structure provided by the frequency of pair-wise interactions. This provides further evidence of abiotic factor influence on facultative mutualism and biotic plant defense.

Habitat heterogeneity in vegetation structure and distribution of EFN-bearing plant species suggest variability in plant strategies for biotic anti-herbivory defense. In our study the plant species in shaded habitats have disperse EFNs more frequently, while plants at open habitats have circumscribed EFNs with most frequency. The latter EFNs are more structurally complex glands (i.e., elevated or pit nectaries) and are more effectively protected against nectar evaporation (Koptur, 1992; Nepi, 2007), which is important at these open, insolated, high-temperature sites.

Non-neutral effects were detected in the ant-plant interacting community since EFN-bearing plant abundance per se had no effects in the ant-plant interaction pattern. As we showed before, more ecological/biological factors, such as habitat/vegetation structure, could affect network structure. Thus, possible convergence effects of interacting species in open vs. shaded habitats could be occurring presided by supergeneralist species and consequently, the possibility of cascading coevolutionary events taking place. This may deserve further study considering ecological/abiotic and coevolutionary contexts for mutualistic interaction networks (Guimarães et al., 2017).

Supplemental Information

Appendix S1 Ant-plant interaction matrix and attributes

Click here for additional data file.

We wish to acknowledge Mariana Cuautle, Rosa Linda Robles, and Gloria Castelazo for their assistance during fieldwork. Fernando Ortega helped with the characterization of extrafloral nectaries. Paulo R. Guimarães and Pedro Jordano made suggestions that improved this manuscript at early stages. Exclusively, CDC thanks for the achievement of this study to God Almighty for the gifts of life, of Christ and his unchanging love and grace.

Additional Information and Declarations

Competing Interests

Author Contributions

Data Availability

The authors declare there are no competing interests.

Cecilia Díaz-Castelazo conceived and designed the experiments, performed the experiments, analyzed the data, prepared figures and/or tables, authored or reviewed drafts of the paper, improved manuscript, and approved the final draft.

Cristian A. Martínez-Adriano analyzed the data, prepared figures and/or tables, authored or reviewed drafts of the paper, improved manuscript, and approved the final draft.

Wesley Dáttilo analyzed the data, authored or reviewed drafts of the paper, improved manuscript, and approved the final draft.

Victor Rico-Gray conceived and designed the experiments, authored or reviewed drafts of the paper, and approved the final draft.

The following information was supplied regarding data availability:

The raw data are available as a Supplementary File.

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
