# Peer review of "Relative contribution of ecological and biological attributes in the fine-grain structure of ant-plant networks"

_PeerJ, doi:10.7717/peerj.8314_

## Round 0.1 · original submission · Major Revisions

The manuscript was reviewed by three external reviewers, all of whom recommended major revisions. In fact all three reviewers had very similar comments that need addressing. While all of the reviewers were impressed by the dataset, they all agreed that the manuscript is difficult to read and interpret. Lack of clarity in the methods made it very difficult to interpret the data and results. Furthermore, Reviewer 2 had particular concerns about the definitions (eg. nestedness) and the analyses selected (modularity). Reviewer 3 had concerns about the appropriateness of the NMDS. Major revisions are required to ensure enough detail is presented in the methods section, terms are well defined, and the readability of the manuscript is drastically improved. Please pay particularly attention to this during revisions, and to the Reviewers’ queries around terminology and analyses.

Reviewer 1 ·

Basic reporting

Manuscript needs work to create a clearer story. Important information left out (relating to variables) required. I.e. Invasive status is used as a variable to explain variation of pairwise interactions, though why this was included as a variable and how it could be important should be explained.

Experimental design

Methods require more detail - no information on how nectar volumes were obtained or how species dominance hieracharchy scores were obtained. Furthermore details lacking on how vegetation along the 6x 1km tracks was observed - all plants?, or restricted to plants of a certain height, surveyed every transect walk? etc.

Validity of the findings

No comment.

Additional comments

Consistent grammatical errors throughout manuscript.

Line 330 - States detailed visitation frequencies recorded in Table 2. Table 2 does not display this information.

Reviewer 2 ·

Basic reporting

This report refers to the manuscript "Relative contribution of ecological and biological attributes in the fine-grain structure of ant-plant networks" by C. Díaz-Castelazo and V. Rico-Gray. The authors investigate a variety of mechanisms potentially shaping the structure of an ant-plant interaction network, particularly with regards to its fine-grain structure. The authors described a nested network and found that habitat heterogeneity is the major factor shaping both the macro and "fine-grain" structures of the network. The work addresses a general question of broad interest for those studying the structure and dynamics of ecological networks. The generality of the research question, combined to a very robust dataset that for tropical ant-plant interactions make this article a potentially relevant contribution. However, I have several concerns regarding the quality of the text. The manuscript has many long and complex sentences, which are often unintelligible. This problem is particularly critical in the Introduction and Discussion sections. It affects the flow of ideas, the conceptual accuracy, and makes it difficult to follow the general reasoning that relates the questions, methods, results and conclusions. As a consequence, it is also difficult to assimilate the take-home messages. In addition, I have a few other concerns about the study design and the validity of the findings, which are detailed later in my report. Therefore, in my opinion, the paper fails to properly communicate its background, results and impacts and, for this reason, cannot be accepted for publication at this point. For these reasons, I have decided to recommend a major revision.

I am a non-native English speaker and, therefore, I am certainly far from being the best person to provide Grammar-related advice to the authors. However, I would like to emphasize that the paper needs a detailed proof-reading to improve the English language usage. In what follows, I indicate other general problems and potential solutions that I hope will help the authors to improve their manuscript.

(1) The Introduction should be rewritten to properly introduce the research questions to a broader audience, which is not necessarily familiar with the jargon, concepts, and theory of network ecology. For example, in the second paragraph of the Introduction (Lines 88-105), the authors list process that arguably account for nestedness in mutualistic networks, without even defining nestedness. Nestedness is only briefly (and incompletely) defined at the 4th line of the Abstract. Please introduce the nestedness concept using a more detailed explanation, referring to works such as Almeida-Neto et al. (2008, Oikos 117: 1227-1239). Also, the list of processes accounting for nestedness is redundant with the processes listed in the first paragraph as drivers of the more general notion of network organization. I believe that the flow of ideas would be more intuitive if the text first state that ecological networks exhibit non-random structural patterns subsequently indicate which structural patterns are most recurrently found in different types of mutualistic networks (e.g., nestedness and modularity) and, finally, introduce the mechanisms that arguably account for each pattern. Another point to be considered in the new version of the Introduction is a more detailed/organized explanation of what is the "fine-grain" structure of the network, and the candidate processes accounting for it.

(2) The text would also benefit from the replacement of several sentences that are too long, too complex, or simply unintelligible. In order to improve the flow of ideas, replace those sentences by shorter and properly concatenated smaller sentences. For instance, you could edit the long/complex sentences listed below. Please notice that this list is not extensive and other long/complex sentences should be rephrased accordingly.
(2.1) Lines 74 - 76. Split the sentences.
(2.2) Lines 89-93. This sentence is central in the Introduction, as it summarizes the mechanisms that can lead to nestedness. However, it is not correctly written. Split and/or rephrase the sentence.
(2.3) 97-101. Split the sentences and provide more details to introduce the concept of "fine-grain structure", which is a key idea for the article.
(2.4) 106-110. Split the sentences.
(2.5) 131-135. Split the sentences.
(2.6) 352-355. Split the sentences.
(2.7) 375-381. Split the sentences.
(2.8) 382-386. The sentence is too long and difficult to follow.
(2.9) 386-391. Split the sentences.
(2.10) 391-395. The sentence is too long and difficult to follow.

(3) Grammatical problems are recurrent throughout the text and should be fixed. One of the most recurrent grammatical problems is the incorrect use of commas at several instances (e.g., at the first line of the Abstract and at Lines 228 and 371). Other types of grammatical issues also occur (e.g., Line 75). Many other grammatical problems appear and, once again, I strongly suggest a detailed proof-reading aiming to improve the English language usage.

(4) The authors use the expression "potentially mutualistic networks" throughout the text, although the reason for that is explained only in the Discussion. Please move such an explanation to the Introduction.

Experimental design

(5) The text would benefit from the addition of equations to define the metrics, such as "species strength" and d' ("specialization of each species").

(6) At line 230, please clarify what you mean by "... a more meaningful measure of network complexity".

(7) The authors state that "... the fact that no compartment existed within the network and its high nested structure render the formal modularity analysis unnecessary" (Lines 225-227). I cannot agree with this statement. Nested networks can hold variable degrees of modularity, and can even be strongly modular (Fortuna et al. 2010, Journal of Animal Ecology 79: 811-817). Therefore, the detection of a nested structure "per se" cannot be used as a justification for not performing modularity analyses. In addition, according to the analysis that I performed using the dataset provided for review and the same software used by the authors (Modular, Marquitti et al. 2014, Ecography 37: 221-224), I detected a low degree of modularity, which is not different from the expected under the null models (Q_B = 0.30, P_Null_1 =0.87, P_Null_2=0.65, after running 100 null models). P-values here are used as a reference to evaluate whether observed modularity depart from the expected under theoretical benchmarks, namely the Erdos-Rényi model (P_Null_1) and the null model 2 of Bascompte et al. (2003, P_Null_2, see Marquitti et al. 2014, Ecography 37: 221-224). Therefore, the authors’ description of this network as "no-modular" (e.g. Line) can be misleading. All instances of "no-modular" and "non-modular" should be properly rephrased. I suggest that the authors perform and report the modularity analyses, as they should not be time-consuming and would improve the characterization of this ant-plant network.

Validity of the findings

(8) A major problem of the Discussion is the lack of a clear delimitation between inferences and generalizations that arise strictly from the paper results and those that are actually references to other studies or speculations. For instance:
(8.1) The discussion opens with the idea that the "super-generalist" species that form the core of the interactions within the nested network are "fundamental components of the maintenance of convergence" (Lines 352-353), which is an idea arising from the theoretical work by Guimarães et al. (2011, Ecology Letters 14: 877-885). However, such idea is not contextualized for the study system, for example, by illustrating what "convergence at the community level" means for this network and how "the addition and persistence of more specialized species" would take place for the case of these ants and plants.
(8.2) As the authors state at different instances that core ant species provide biotic defenses to the plants, I would like to see more references supporting the discussion of this point, which is not directly addressed by the study. A strong feature of this study system is the availability of natural history knowledge for ant species. I feel that a more in-depth discussion of this aspect would greatly improve the paper, as the authors are probably quite familiar with the studied myrmecofauna. For instance, the role of Camponotus species as major players in the network core is clearly relevant... but what about the rest of the rest of the ant species? What features of Azteca sp. 1 and Paratrechina longicornis make them core species? In addition, is it possible to think of generalizations relating social features of ant species and the fine-grain structure of this type of ant-plant mutualistic network?

(9) At Line 357, please clarify what you mean by '... plants "depend" more on ants for maintaining network structure...'

(10) Although the authors properly refer to the seminal work of Vázquez et al. (2009) as a key reference to discuss the processes driving the structure of mutualistic networks, I suggest that they also incorporate ideas from more recent works discussing the processes shaping ecological networks, such as Poisot et al. (2015, Oikos 124: 243-251).

Reviewer 3 ·

Basic reporting

The article is, in general, quite difficult to read. The authors have used extremely long sentences (one covering 15 lines!) and this tends of obscure their points. Moreover, some seemingly important points are vague (e.g. lines 94-100) and difficult to interpret. There is also a fair amount of repetition. For example, lines 131-137 in the introduction appear almost verbatim as lines 473-479 in the conclusion, and lines 76-79 are extremely similar to the abstract. The whole article would benefit from a change to clearer, simpler, and more concise sentence structures. Such a revision would also likely help the authors to spot and remove repetitions.

Aside from these general issues of readability, several areas of the study need greater clarification and/or support. I will give these line-by-line:

Lines 85-87: The statements that species with higher interaction frequencies have A) higher numbers of links (does this mean degree?), B) possibly higher abundance, and C) are potentially better competitors, need more support. I would like to see at least one citation per claim. The relationship between interaction frequency and degree especially is counterintuitive since I would expect species to specialize on a limited number of highly profitable links rather than continually adding links as interaction frequency increases.

Lines 92-93: Please give some details on how forbidden links could affect nestedness.

Line 116: What are the "important effects" of extrafloral nectaries on network structure? What aspects of structure do you mean?

Lines 118-119: What's the difference between "network-level" and "community-level"? To me these are the same.

Line 127: Please give some more detail about the social recruitment behavior of ants for those of us without strong entomological backgrounds.

Lines 165-169: It's not clear how you're planning to define network structure. I think it would make your manuscript much clearer to introduce your definition of structure earlier. From the rest of the introduction, I'd expect nestedness to be a major focus but this does not seem to be the case.

Line 308: Please add more details on vegetation associations to the methods. At the very least, I would like a table making explicit which vegetation types are "open" and "closed".

Line 323: It's a little confusing to present the second NMDS axis first.

Lines 369-370: It's very unclear what this means.

Line 388: What is a "high foliar rate"?

Line 427 and elsewhere: You mention that seasonality is important, but I didn't see anything in your analysis to back this up. More details needed.

Line 444: Abundance is also missing from your analyses.

Line 503: If you want to end with the possibility of coevolutionary cascades, it would be helpful to add some context for this to the introduction.

Experimental design

The raw data used in this study represents a phenomenal amount of work and will be a rich addition to the field of network ecology. However, there is a lot of room to improve the presentation of the analyses used on this excellent data.

The authors mention seasonality and abundance in their discussion, but not in the methods and results. These should either be removed from the discussion or explicitly added to the methods and results.

The authors' decision to remove infrequent interactions also needs more justification. Weak links can be important in maintaining community stability (for example), so they may be important. Removing these links is also very likely to change the network structure. Perhaps the results for similar analyses including all links could be provided in an appendix.

It is also not clear exactly which measures of network structure the authors are interested in and why. The introduction is largely focused on nestedness, but it seems that the authors then included all available network measures provided by Bipartite. It would help readers to follow the results and discussion if a more thorough definition (and some indication of its importance) was provided for each network structural property.

Validity of the findings

It is not clear whether the authors' claims about the importance of abundance and seasonality are speculation or not. It is difficult to judge the validity of the other results based on the NMDS figure provided. A table or plot of network structural properties over space/time would be very helpful (the authors could highlight a few properties of special interest). In the NMDS plot, some visual differentiation between plant and ant species would aid interpretation.

Additional comments

The authors have created an excellent dataset, but their article as it currently stands is difficult to read and interpret. If the authors can clarify their work, adding more details where appropriate, I am confident that this will be a valuable contribution to network ecology.

---

## Round 0.2 · Major Revisions

While the manuscript has been improved, unfortunately more clarity is required for it to be acceptable for publication. Major revisions are still required to ensure enough detail is presented in the methods section, terms are well defined, and the readability of the manuscript is drastically improved. The reviewer has put an enormous amount of effort into improving the manuscript - we hope this helps you taking the paper forward.

Reviewer 3 ·

Basic reporting

The quality of the writing is greatly improved relative to the previous version, but the article is still confusing in places. The authors still tend to use very complex sentences with nested clauses and many colons, semi-colons and commas; these complex structures mean that any small grammatical errors or unusual word choices take a great deal of reader energy to untangle. I suggest that the authors continue to simplify their writing, particularly in the discussion which is noticeably less polished than the rest of the manuscript. I have identified many small grammatical points (listed at the end of this review) that the authors can quickly address to improve the manuscript. In the same list I identify a few points in the discussion that are confusingly written and need revision.

Most of the authors claims are sufficiently referenced. Only line 601 seems to reference previous work that is not explicitly identified. The R statistical software and vegan package, however, should be cited each time they are referenced (the R base package was not cited at all). This discounts the vast amount of work that went into developing these publicly-available resources and is quite discourteous.

Raw data were shared, but do not include sufficient information to replicate the study. In particular, there is no key to species names or attributes (either the identity of the attribute or the factor levels). It is impossible for me to say which plants appear in which forest types or what types of nectaries they bear. A metadata file explaining the raw data should be provided. Ant head lengths, and abundances were not provided.

While not necessary to replicate the analysis in this paper, to provide the highest value raw data for future re-use it would also help if the raw data included the number of surveys on which an interaction was observed as well as the overall interaction frequency. Ideally, the network provided by each survey would be provided as a unique sample of the overall network. This would provide important information on the variance in interactions over time as well as the variance in frequencies the authors address here.

The manuscript is self-contained and the results reflect the hypotheses.

Detailed comments:
Abstract
-line 18: "showned" should be "shown", "species" should be "species'", and "counterpart" should be "counterparts"
-line 34: should be "highly-nested and non-modular"
-line 36: "higher dependences on ants" seems like it needs "than vice versa"
-line 39: "plants species" should be "plant species"
-lines 39-42: sentence is not grammatically correct. I suggest splitting it into several.
-lines 42-44: How does spatialtemporal heterogeneity depict the effect of abiotic factors? Heterogeneity in what? This sentence is very unclear.

Introduction
-line 50 and elsewhere : should be "in a community" not "at a community"
-line 60: should be "species' dependence on their counterparts"
-line 62: mismatched quotes around "nested"
-line 75: first word should be "Another"
-lines 79-81: This sentence is confusing. Something like "Like nestedness in mutualist networks, modularity is thought to increase the persistence of species in antagonistic networks." It also seems like the sentence starting in line 81 should begin a new paragraph.
-line 98: typo in reference
-line 104: suggest replacing "importantly" with "significantly". Odd word choice.
-line 106: start a new sentence after the reference. Missing a "to" in "refer to here".
-line 110: should be "structuring the fine-scale patterns"
-line 123: "its" should be "their" since networks is plural
-line 130: no hyphen in "network level"
-line 140: should read "include the social..." since you are about to give a partial list of attributes
-line 147: no comma after "behavior"
-line 148: "influence" should be "influences" and "given that" should be "since"
-line 156: remove "-either addressing a couple of biological or abiotic factors,". Not necessary since you state that abiotic and biotic factors were explored separately in line 155.
-line 164: "Which" should probably be "What"

Materials and Methods
-lines 197-199: this ordering is confusing. Maybe "between "open" and "shaded" habitats, with the first three habitats () being included in the former physiognomy and the other three habitats () included in the latter".
-lines 200-201: Define "floristic similarity". How is it calculated? What aspects of flowers are included?
-line 210: reagent stripes or reagent strips?
-lines 220-221: remove commas. "disperse" should be "dispersed"
-line 224: should be "a widely dispersed location and that this could result"
lines 231-242 and 244-256: These two sentences span over 10 lines each and are quite difficult to parse. It may be better to list the attribtes briefly first and then, in separate sentences, give the details about each attribute.
-line 294: "values up to 1" is confusing given that values >1 are reported in the results. Do you mean "values close to 1"?
-line 299: repeated ")."
-line 314: should be "Bray-Curtis dissimilarity"
-lines 320, 323, 324: missing citations for the R base package and a citation for vegan on the second reference
-line 331: the "2" should be a superscript
-lines 331-332: it isn't clear how floristic similarity relates to network structure, especially since the raw data includes only a single network.

Results
-line 339 and elsewhere: should be "consisted of"
-line 340: It isn't clear what the difference is between "quantitative interactions" and "species associations". I infer that quantitative interactions are single observations and species associations are plant-ant pairs, but I'm not sure.
-line 342: should be "detected in"
-line 352: should be "Regarding 'species strength'"
-line 353 and elsewhere: "standout" should be "stand out". "standout" is an adjective and "stand out" is the verb.
- lines 353-355 would be clearer as 2 sentences.
- line 355 and elsewhere: it looks like "EFnectar" should be two words
- line 358: "peculiar" is an unusual word choice. Peculiar in what way? Are these ants morphologically or behaviorally distinct?
-line 366 and elsewhere: "disperse nectaries" should probably be "dispersed nectaries" to match "circumscribed nectaries".
-line 367: "amount of" should probably be "number of"
-line 375: it seems like a word is missing here. Maybe "by attracting a distinct group of associated ant species"?
-line 377: "considerable" should be "considerably"
-lines 382-383: "being an acceptble value" doesn't make much sense. I suggest shortening this line to "... (0.17) and suggests that ..."
-line 390 and elsewhere: "cero" should be "zero"
-line 392: "superior part" should be "upper part"
-line 408: "were significant to explain" doesn't make sense. You could simply say "explained". "network fine-grained structure" should be "network's fine-grained structure"
-line 413: "differential associations form" should be "different associations from"

Discussion
-line 421: "occur" should be "occurs"
-line 422: "relative" should be "relatively"
-line 423: "exist" should be "exists"
-line 424: "components of" should be "components for"
-line 428: "to forage" should be "for foraging"
-line 442: remove "mere", "explain" should be "explained"
-line 444: "possible" should be "possibly"
-line 448: again, "peculiar" is an odd word and makes me think that there is something wrong with the ants the interact with highly-specialised plants.
-line 460: "of rapid up taking" should be "to rapidly take up"
-line 474: subheading is not grammatically corred. Maybe "Relative contributions of attributes to the assemblage of pairwise interactions"?
-line 475: remove comma. Seems to be missing a word - maybe "associations grouped according to habitat"?
-line 477: "seems" should be "seem"
-line 490: "carbohydrates" should be "carbohydrate"
-lines 491, 498, 505, 508, 510: remove commas
-line 508: should be "among-habitat" and "provided in"
-line 510: "display" should be "displays"
-lines 512-516: This sentence makes me miss a brief description of the network as well, just to tie it all together.
-line 519: should be "considered in our analysis but rendered"
-line 521: remove comma
-line 525: should be "more plant species with EFNs"
-line 527: "being" should be "was"
-lines 527-528: the clause after the comma doesn't clearly relate to the first clause.
- lines 528-529: This isn't clearly shown... did you also find this in your study? Wasn't clear in the results.
-line 530: "Other potentially..." seems like it could start a new paragraph.
-line 538-539: it's not clear whether you mean that the intensity and duration of your sampling could demonstrate the spatial and temporal variation, or that it was not sufficient to do so? As written, it could be either, and the raw data are a single web so give no information about any sort of variation.
-line 543: should be "more thoroughly explains"
-lines 546-549: I suppose - but since your study did not show any such effect this does not seem particularly interesting.
-lines 549-553: This is the first time plant defense platicity has been mentioned, and it seems to come from out of nowhere. The use of "plasticity" makes me think of developmental flexibility but seems to be used as a synonym for "variability" here. It is also not immediately clear how EFNs relate to plant defense. I assume the nectaries attract ants which do the actual defending, but it is probably worth briefly stating this so that your paper is clear to network ecologists who do not work with ants. There aren't enough details in the data and results to let me evaluate this claim for myself (I can't find which plants have EFNs or whether they have more diverse sets of ant partners).
-line 557: Order confusing. "attributes, such as species degree, within a mutualistic network" would make more sense.
-lines 555-563: This paragraph seems to be saying that bigger ants tend to lose competitions and therefore need to forage more widely. That is very counterintuitive.
-line 564: Add comma after "Overall"
-lines 566-567: remove commas.
-lines 564-568: Very long. Split into two sentences.
-line 572: "seems in accordance to" is a very odd phrase to have here. "makes sense in" or something similar would be better as it does not invoke any sort of laws/rules in human-altered systems (that have not been mentioned)
-line 574: "do" should be "does"

Conclusions
-line 581: "result" should be "was"
-line 582: "show" should be "showed"
-lines 583-586: not grammatically correct and confusing. Ending with plant defense and ant foraging adaptations pins this sentence on the weakest, least-supported part of the discussion and undermines your point.
-line 591: "in network topology" and "in the fine-grain network structure"
-line 592: "being thus further evidence" is strangely legalese phrasing and a bit confusing. I suggest ending the sentence at "interactions" and starting a new one with "This provides".
-lines 594-599: Are these things common knowledge? They were not clearly demonstrated in the present study.
-line 598: Add "and are" before "more effectively" to make this grammatically correct.
-line 601: Add a citation for previous work where you show that other factors could affect network structure. Only habitat type seems to have an effect in the present study.

Figure legends
- Figure 1: It's a bit confusing to have core ants and plants both be blue. Maybe dark/light red and green to separate core/peripheral species would work.
- Figure 2: lines 870-871 seem copied from the results and have the same English errors. This is not a complete sentence.
Line 873: What does "Distant ellipses" mean? They seem to be in the middle of the plot and not particularly distant from anything. How was it determined which attributes explained the pattern? No attributes are presented on the plot, so it is impossible to check this. It would also help to include your interpretations of the axes in the figure legend.

Experimental design

This original research fits the journal aims and scope.

The research question is now well-defined and meaningful, and the introduction now clearly establishes the knowledge gap the authors intend to fill.

The experimental design is generally well described and rigorous, with the exception of analyses relating to abundance. It is not clear how plant and ant abundances were measured (were they counted using the honey-bait traps that were briefly mentioned in line 202? If so, it is not clear that this information was used in the analyses.) and whether they were explicitly controlled for in the analyses. No supporting data are provided, for example, that would let me see whether abundance relates to the positions of species on the NMDS plot. If ant and plant abundances were not measured independently of interaction frequencies, then it is not possible to make strong claims about abundance as abundance and interaction frequency may or may not be correlated (as the authors point out). If abundances were sampled independently, this information should be provided with the raw data. It is also not clear how floristic similarity was calculated.

The methods lack key details (abundance, floristic similarity) and the raw data are missing some elements that make it impossible to repeat the analyses used in this study. The study would be difficult to properly replicate without more information on how abundances were determined and what, exactly, floristic similarity means.

Validity of the findings

The association between different plants and habitat groups/structures is not clear enough to properly assess the conclusions. A table summarizing ant and plant attributes would be extremely helpful in interpreting Fig. 2 and would substantially strengthen the manuscript. It is currently impossible for a reader without expert knowledge of the system to judge the authors' claims about which traits structure the community. None of these claims are particularly unbelievable, so I am confident that the authors can support them with a more detailed presentation of their evidence.

Some of the conclusions are well-supported by the rest of the manuscript but the authors' claims about plasticity of defensive strategies are not. This may be speculation, but it is not clearly identified as such. Further revisions to the discussion may help to solve this problem.

Additional comments

I respect the work that went into assembling this dataset, but the analyses remain opaque enough that I am not confident in the conclusions presented. In particular, the theme of plasticity in plant defense appears out of nowhere in the discussion and does not clearly connect to the results. This is an interesting idea, so I suggest that the authors add plant defense to the introduction and make sure it is a clear thread through the entire manuscript.

---

## Round 0.3 · Major Revisions

While the science is much improved in this revision - unfortunately the English is not up to standard for the journal. I've spent quite a bit of time going through it (see the attached document) - but it needs a thorough re-write to be acceptable. I suggest you pay for a service if you have no one who can do this for you.

The article also seems far too long. There are several places where it could be drastically reduced - mainly via removing repetition and re-writing. The introduction in particular is far too long - it takes a long time to get to the point of the manuscript. Shorten please e.g. The first two pages of the introduction could be summarised and reduced into half a page. I've pointed out other places where it could be reduced within the manuscript.

Figure 2. Rather than having ant and plant names on nMDS - which are difficult to read – could you better explain the factors and habitat types show on the nMDS. Its not very self-explanatory at the moment.

---

## Round 0.4 · Minor Revisions

As noted by the reviewer (and I read the paper as well) the language is much improved. However there are a few small language issues which, if fixed, would make this paper even better.

In addition, please pay attention to the other comments from the reviewer – e.g. information on identification of plants and ants, and information on data deposition (as per PeerJ requirements).

Please provide a detailed response to the issues brought up by the reviewer.

Reviewer 3 ·

Basic reporting

The English is still not perfect (see list of corrections in general comments) but is generally clear. There are only a few ambiguous sections (see list of corrections). The article is professionally structured, adequately referenced, and self-contained.

Experimental design

I have no further comments on the experimental design.

Validity of the findings

The findings seem to be valid and well-supported.

Additional comments

The authors have further improved the writing of the manuscript and it is now mostly clear, although a large number of small language errors and some run-on sentences remain, as well as a few places where a little more information would be helpful. These should be quite easy for the authors to fix, leaving quite a nice manuscript relating plant-ant network structure to habitat structure. However, I am still concerned that the data provided by the authors does not include all of the information that would be needed to re-create the study. There is still no key to plant and ant names, only abbreviated trait names are given in the "Attributes" tab, and I do not see ant head lengths, etc. Please expand the descriptions in the data to meet PeerJ's standard for raw data publication


Minor comments:

line 224: what was the minimum time before you determined that feeding had occurred?
line 252: it would help readers who are not ant specialists to briefly explain the difference between invasive and tramp species, if there is space.
line 288: NODF is less sensitive than what? I assume some other measurement of nestedness, but it would be good to be specific.
line 363: what was the value for Barber's Qn?
line 386: why don't you give d' for ants?
lines 530-539: these results only mention species strength. Did you examine the relationship between species strength and habitat?
lines 533-535: In the results it seemed like this was only the upper part of axis 2, not an overall trend. You may want to adjust the results or discussion so that they are consistent.
lines 571-572: by "simultaneous contribution", do you mean that each variable was significant on its own? This is somewhat confusing.
Table 2: Are these dominance categories hierarchical (i.e., A beats B, B beats C)?

Language comments:

Abstract

line 41: should read "species which, together with higher dependence of plants on ants, suggests potential...
line 43: "so prevailing that" should read "so prevalent that it"


Introduction

line 108: "evidence" should be "demonstrate". Evidence is only a noun in English, not a verb.
line 128: "render" should be "result in" or maybe "yield"
line 129: should read "plenty of variation in EFN..."
lines 205-209: This sentence is long and hard to parse. Can you break it up into a few shorter sentences?

Methods

lines 261-270: This sentence is extremely long and hard to read. I strongly suggest that you break it up into shorter sentences. Lines 267-270 are especially hard to follow.
lines 290-291: It is very helpful of you to include the names of the null models in two papers! That should make things much easier for your readers.

Results

line 364: "exist at" should read "exist in"
lines 367-368: "that is network still" should read "that the network is still"
line 373: "Regarding species strength" is a somewhat strange beginning to a sentence, and should probably be followed with a :
line 394: "within" should probably be "to have"
line 401: You give F=15.80 and F=15.79. This is a very small difference, but one of these values is probably wrong.
line 407: "having as well" should be "also had"
line 412? remove comma
line 433: "at our study" should be "in our study"

Discussion

line 442: should be "structure and is thus..."
lines 441-444: run-on sentence. Break into two or more. It is also not clear whether you mean that modules are a common feature of simulated networks or that the lack of modules is a common feature.
line 448: "community-level" should be "community level"
line 462: "at a community level" should be "at the community level"
line 473: "accordance to" should be "accordance with"
line 478: "species that constituting" should be "species that constitute"
line 486: "in nectar" should be "on nectar"
lines 486-492: run-on sentence. Break up.
line 492: "At our studied" should be "In our studied"
lines 492-495: I'm not sure what these lines are supposed to be saying. Do only the core ants provide biotic defence? If not, why are they specifically pointed out?
line 530: should be "outstanding as"
line 531: "transcending" should be "translating"
line 546: "At our study" should be "In our study"
lines 546-547: "in order to explore its effect in network structure" makes it sound like you excluded abundance in order to test its effect, which does not make sense. You could just remove this text.
line 561: "to account for interactions" might be more accurate as "to account for variation in interactions"
line 563: "capture" should be "captures" since you are talking about one set of censuses
line 583: "at plant species" should be "of plant species"
line 590: remove comma. It is also not clear whether you mean these plant-ant mutualistic networks, or some different ones.
lines 593-597: run-on sentence. Break up.
line 594: "resource in" should be "resource is"
line 599: "spectra" should be "spectrum"
line 603: "seems" should be "seem"
lines 607-610: I don't think this needs to be a new paragraph.

Conclusion

lines 627-628: Why did you change from EFN to EFnectaries here?
lines 946-947: "having as well" should be "also has". This text is also a direct copy of that in the main text.

---

## Round 0.5 · accepted · Accept

Thank you to the reviewers for their work on this manuscript. Particular thanks to reviewer #3 who worked with the co-authors over three iterations. And thanks to the co-authors for their clear responses and rebuttals.

I believe, following this review process, that this paper is now ready for publication in PeerJ. Congratulations, and thank you for choosing our journal to showcase your work.